# Increasing *in vivo* drug exposure levels of compound WX-081 (sudapyridine) when used in combination with clofazimine or clarithromycin

Xueyu Wang,[1] Shan Gao,[1,2] Xia Yu,[3] Yongguo Li,[4] Lei Li,[4] Naihui Chu,[1] Hairong Huang,[3] Wenjuan Nie[1]

**ABSTRACT** To examine the *in vivo* distribution of WX-081 alone or in combination with clarithromycin (Clr) or clofazimine (CFZ) in rats infected with *Mycobacterium abscessus* and evaluate the effects of the interactions of WX-081 with Clr and CFZ on its tissue distribution, the concentrations of WX-081 in plasma, brain, vertebral, and lung tissue of Sprague-Dawley rats at 20 min, 1 h, and 16 h post-treatment with 45 mg/kg WX-081 (Group A), 45 mg/kg WX-081 plus 10 mg/kg Clr (Group B), or 45 mg/kg WX-081 plus 25 mg/kg CFZ (Group C) were determined by liquid chromatography-tandem mass spectrometry. Differences between continuous variables were analyzed using Student's *t*-test ($P \leq 0.05$). At 20 min, rats in groups B and C had higher plasma WX-081 concentration than those in Group A (458.7 ng/mL vs. 140.4 ng/mL, $P = 0.012$; 351.7 ng/mL vs. 140.4 ng/mL, $P = 0.022$, respectively). At 1 h, only Group B had a higher plasma level than Group A (2,522.3 ng/mL vs. 1,413.2 ng/mL, respectively, $P = 0.018$), while the WX-081 concentration in Group B lung tissue exceeded that in Group A lung tissue (8,890.9 ng/g vs. 6,666.8 ng/g, respectively, $P = 0.041$), and that of Group C lungs was lower than that of Group A lung tissue (3,953.4 ng/g vs. 6,666.8 ng/g, respectively, $P = 0.014$). The lung tissue consistently had the highest WX-081 concentration at all time points. At 20 min, plasma WX-081 levels in groups B and C surpassed that of Group A. At 1 h, Group B had higher plasma and lung WX-081 concentration, while Group C had a lower lung WX-081 concentration than Group A.

**IMPORTANCE** The study pioneers the exploration of WX-081's *in vivo* distribution and drug interactions with Clr or CFZ in a rat *Mycobacterium abscessus* infection model. It shows that co-administration alters tissue distribution, boosting lung concentrations—key for treating pulmonary infections. These insights guide regimen optimization and underscore WX-081's potential as a safer alternative, enhancing treatment for non-tuberculous mycobacterial diseases.

**KEYWORDS** WX-081, clarithromycin, clofazimine, drug distribution, *Mycobacterium abscessus*

Global incidence rates of nontuberculous mycobacterial (NTM) lung disease and associated mortality are steadily increasing (1, 2). This is partly due to the innate resistance of NTM to most antibiotics. However, macrolides, such as clarithromycin (Clr), have long been effective treatments for NTM infections. This has led to their inclusion in multidrug regimens for common pulmonary NTM diseases caused by *Mycobacterium avium* complex (MAC) and *Mycobacterium abscessus* complex (MABC). In fact, these drugs are recommended by the current treatment guidelines established by the World Health Organization (WHO) (3). Clofazimine (CFZ), a WHO-recommended Group B riminophenazine antibiotic for drug-resistant tuberculosis (TB) (4), is also included in treatment

**Peer Reviewer** Hao Li, China Agricultural University, Beijing, China

Address correspondence to Wenjuan Nie, wenjuan.nie@outlook.com, Naihui Chu, chunaihui1994@sina.com, or Hairong Huang, huanghairong@tb123.org.

Xueyu Wang, Shan Gao, and Xia Yu contributed equally to this article. Author order was determined based on consensus reached through negotiation among team members.

The authors declare no conflict of interest.

See the funding table on p. 10.

regimens for *M. abscessus* infections. It shows promise as a potential treatment for MAC infections (3).

Bedaquiline (BDQ) is a diarylquinoline antimycobacterial drug. It is used in combination with other antibacterials to treat pulmonary multidrug-resistant TB. It suppresses mycobacterial growth by inhibiting ATP synthase, an enzyme crucial for ATP synthesis (5). BDQ has also demonstrated effective antimycobacterial activity against NTM (5). However, its administration can cause adverse effects, such as cardiotoxicity (prolongation of the QT interval), which have limited its clinical use (6). This has prompted researchers to seek safer antimycobacterial drugs.

One such candidate is WX-081 (sudapyridine), a recently identified diarylpyridine analog that is metabolized in rats to produce the metabolite WX-081-M3 (7). WX-081 has antimycobacterial activity against *Mycobacterium tuberculosis* resembling that of BDQ (8) and anti-NTM activity *in vitro* (9). Moreover, it is a potentially safer drug than BDQ (7, 8). WX-081 is currently undergoing evaluation as a potential TB treatment in a Phase III clinical trial in China (NCT05824871).

To better understand the *in vivo* distribution of WX-081, we conducted this preliminary study. We used a rat model to determine WX-081 accumulation in various rat tissues when administered alone or in combination with other anti-NTM drugs. In addition, we assessed effects of combining WX-081 with either Clr or CFZ on its *in vivo* distribution. This helped us identify antagonistic and synergistic drug interactions.

## RESULTS

### Detection of WX-081 and WX-081 M3 prototype substances

WX-081 and WX-081-M3 were successfully separated by liquid chromatography-tandem mass spectrometry (LC-MS/MS) without interference from endogenous compounds (Fig. S1). Wide linear ranges of detection were observed for plasma, vertebral, and lung samples (50–1,000 ng/mL) and brain samples (10–1,000 ng/mL). All these ranges were with Pearson correlation coefficients ($r$) of ≥0.99. Table S1 provides a precise and detailed description of the procedure for determining the concentration of WX-081 and WX-081-M3 in plasma and tissue by LC-MS/MS. Table S2 shows the WC-081 standard curves and linear detection ranges for all four sample types.

### Distribution of WX-081 and WX-081-M3 in rat plasma and various tissues

After treating Sprague-Dawley (SD) rats with different regimens, specifically groups A (45 mg/kg WX-081), B (45 mg/kg WX-081 plus 10 mg/kg Clr), and C (45 mg/kg WX-081 plus 25 mg/kg CFZ), we determined the levels of WX-081 in plasma, lung, vertebrae, and brain. These samples were collected at 20 min, 1 h, and 16 h post-treatment using LC-MS/MS. The analysis found the highest concentration of WX-081 in the lung tissue of all three groups at all three time points. This was followed in decreasing order by plasma, vertebrae, and brain, as shown in Fig. S2. The concentrations of WX-081 and WX-081-M3 for all samples are listed in Table 1.

### Intergroup comparisons of average blood plasma WX-081 and WX-081-M3 concentrations

At 20 min, the average plasma concentrations of WX-081 in groups B and C were higher than that in Group A (458.7 ng/mL vs. 140.4 ng/mL, $P = 0.012$; 351.7 ng/mL vs. 140.4 ng/mL, $P = 0.022$, respectively). At 1 h, only that in Group B was higher than that in Group A (2,522.3 ng/mL vs. 1,413.2 ng/mL, $P = 0.022$). At 16 h, the average plasma concentrations of WX-081 in groups A, B, and C were 2,402.1, 2,834.2, and 3,198.4 ng/mL, respectively. No statistically significant intergroup differences were observed. The WX-081-M3 metabolite was present at detectable concentrations in plasma of all three groups at 16 h, with no significant intergroup differences (Fig. 1). However, the metabolite was undetectable in all groups at the other time points.

**TABLE 1** Concentrations of the drugs WX-081 and WX-081-M3 in rat plasma and other tissues[a]

| Compound | Group | Plasma (mean, ng/mL) 20 min | 1 h | 16 h | Brain tissue (mean, ng/g) 20 min | 1 h | 16 h | Vertebral tissue (mean, ng/g) 20 min | 1 h | 16 h | Lung tissue (mean, ng/g) 20 min | 1 h | 16 h |
|---|---|---|---|---|---|---|---|---|---|---|---|---|---|
| WX-081 | A | 140.4 | 1,413.2 | 2,402.1 | – | – | 774.8 | – | 1,729.0 | 9,427.9 | 663.1 | 6,666.8 | 26,781.0 |
| | B | 458.7[b] | 2,522.3[b] | 2,834.2 | – | – | 335.8 | – | 1,493.8 | 13,148.6 | 1,184.6 | 8,890.9[b] | 40,560.0 |
| | C | 351.7[b] | 1,231.5 | 3,198.4 | – | – | 246.0 | – | 614.6 | 5,802.4 | 1,290.6 | 3,953.4[b] | 29,017.5 |
| WX-081-M3 | A | – | – | 303.4 | – | – | 201.2 | – | – | 1,970.3 | – | 758.5 | 33,139.6 |
| | B | – | – | 238.9 | – | – | 136.5 | – | – | 1,747.8 | – | – | 34,697.3 |
| | C | – | – | 225.5 | – | – | – | – | – | 1,341.1 | – | – | 25,208.4 |

[a]–, undetectable.
[b]Comparison to Group A, $P < 0.05$.

## Intergroup comparisons of concentrations of WX-081 and its metabolite WX-081 in the brain tissue

As depicted in Fig. 2, WX-081 was detectable only in the brain tissue at 16 h post-treatment. The highest concentration of WX-081 was observed in the brain of Group A (774.8 ng/g). However, there was no significant difference between groups A and B ($P = 0.379$) or between groups A and C ($P = 0.291$). WX-081-M3 was detected only in the brain of groups A and B at 16 h.

## Intergroup comparisons of concentrations of WX-081 and its metabolite WX-081 in vertebral tissue

As shown in Fig. 3, although WX-081 and its metabolite were not detectable in rat vertebrae at 20 min, measurable amounts of WX-081 were detected at 1 h in groups A, B, and C. The concentrations were 1,729.0, 1,493.8, and 614.6 ng/g, respectively. At 1 h, WX-081-M3 was not detectable in vertebral samples from any of the rat groups. But at 16 h, it was detected in vertebral samples from groups A, B, and C, with concentrations of 9,427.9, 13,148.6, and 5,802.4 ng/g, respectively. No significant difference was observed between groups A and B ($P = 0.390$) or between groups A and C ($P = 0.223$).

## Intergroup comparisons of concentrations of WX-081 and its metabolite WX-081 in lung tissue

As illustrated in Fig. 4, at 20 min, WX-081 concentrations in lung samples were all higher than the corresponding plasma concentrations of WX-081 at 1 h. Also, at 1 h, the concentration of WX-081 in Group B was significantly higher than that in Group A (8,890.9 ng/g vs. 6,666.8 ng/g, $P = 0.041$). In contrast, the concentration in Group C was lower than that in Group A (3,953.4 ng/g vs. 6,666.8 ng/g, $P = 0.014$). Its metabolite WX-081-M3 was detectable only in the lung tissue from Group A. At 16 h, the concentrations of WX-081 in the lung tissue from all three groups were significantly higher than those at 1 h. Moreover, the concentrations of WX-081-M3 were measurable for the first time in groups B and C.

## DISCUSSION

WX-081 is currently in the clinical trial phase in China. As a result, it has not been widely administered to patients in clinical settings. This compound, also known as sudapyridine, is a novel diarylpyridine analog. Its molecular formula is $C_{34}H_{33}ClN_2O_2$. It is synthesized by substituting the bromoquinoline group of BDQ with a 5-phenylpyridine. Its metabolite, WX-081-M3, has a molecular formula of $C_{33}H_{31}ClN_2O_2$. Notably, Huang et al. (7) demonstrated that WX-081 accumulates in the lung tissue. This is consistent with the current study's findings, indicating that the highest concentration of WX-081 is found in the lung tissue compared to other tissues. Thus, WX-081 holds promise as a treatment for pulmonary mycobacterial infections. Yao R et al. (8) conducted comprehensive analyses

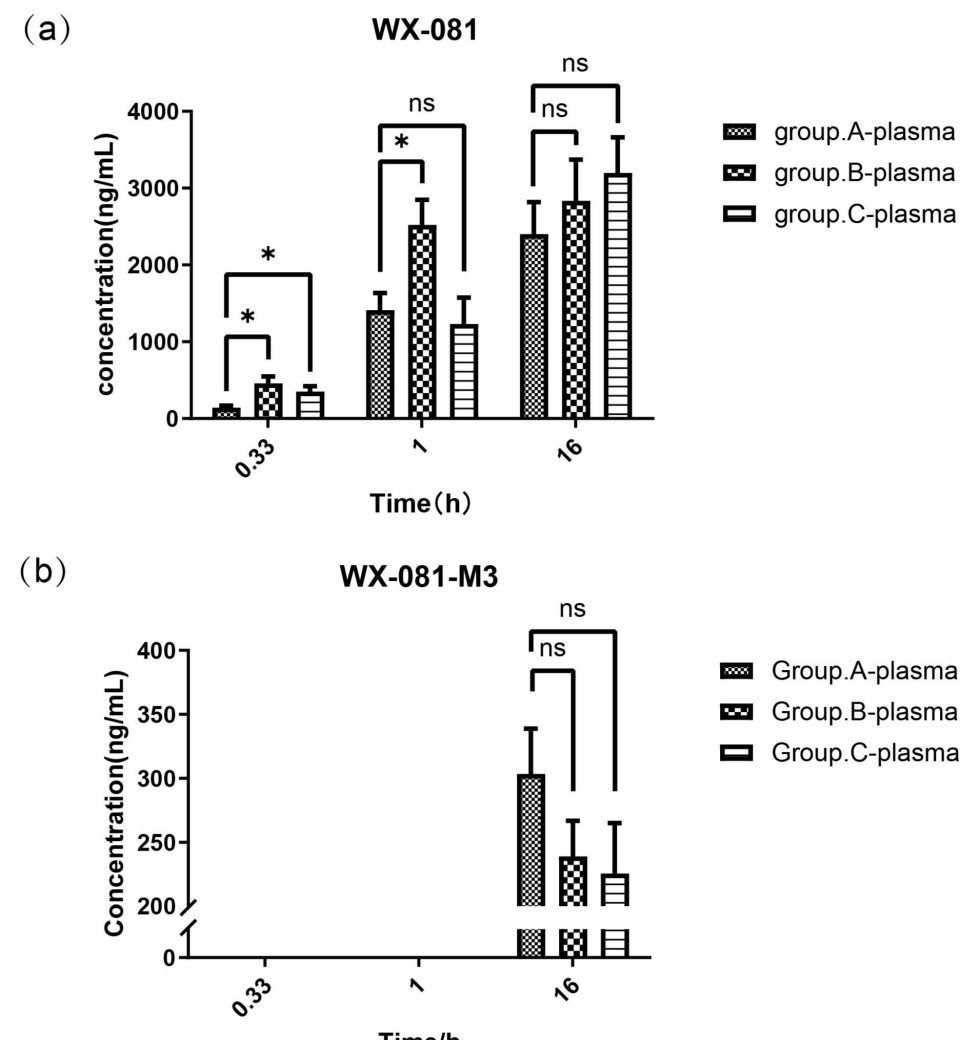

**FIG 1** Concentrations of WX-081 (a) and WX-081-M3 (b) in plasma from groups A, B, and C at three time points. ns: not significant; *: $P < 0.05$.

on the pharmacokinetic parameters of WX-081. In beagle dogs, oral doses of 2, 6, or 20 mg/kg yielded the time to reach peak concentration ($T_{max}$) of 4.6–7.7 h, the maximum plasma concentration ($C_{max}$) of 390–1,660 ng/mL, and the area under the curve from time zero to infinity ($AUC_{0-inf}$) of 9,490–58,200 ng·h/mL, showing dose-dependent exposure. Intravenous volume of distribution (Vd) in mice and rats was 10.4 and 9.2 L/kg, respectively, indicating good tissue penetration. The clearance (Cl) was 3.59 mL/min/kg in mice and 8.25 mL/min/kg in rats, suggesting moderate clearance. The elimination half-life ($t_{1/2}$) in beagle dogs ranged from 51 to 58 h, favorable for sustained therapeutic levels.

To date, few studies have examined whether WX-081 can cross the blood-brain barrier. However, our previous study (10) revealed that WX-081 had an antimicrobial effect on mycobacteria in the brain of *M. abscessus*-infected zebrafish. This suggests that WX-081 crosses the blood-brain barrier. In this study, both WX-081 and its metabolite WX-081-M3 were detected in the brain tissue of SD rats. This provides further evidence that WX-081 can cross the blood-brain barrier. However, whether WX-081 can exert an antimicrobial effect in the brain *in vivo* must await further confirmation through additional studies before the drug can be used as an anti-NTM treatment in clinical practice. Remarkably, the detection of WX-081 and its metabolite in the rat vertebral

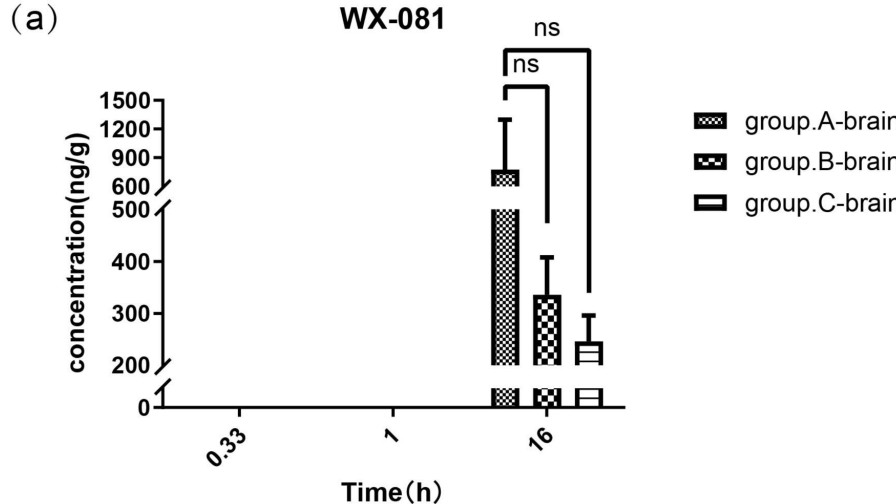

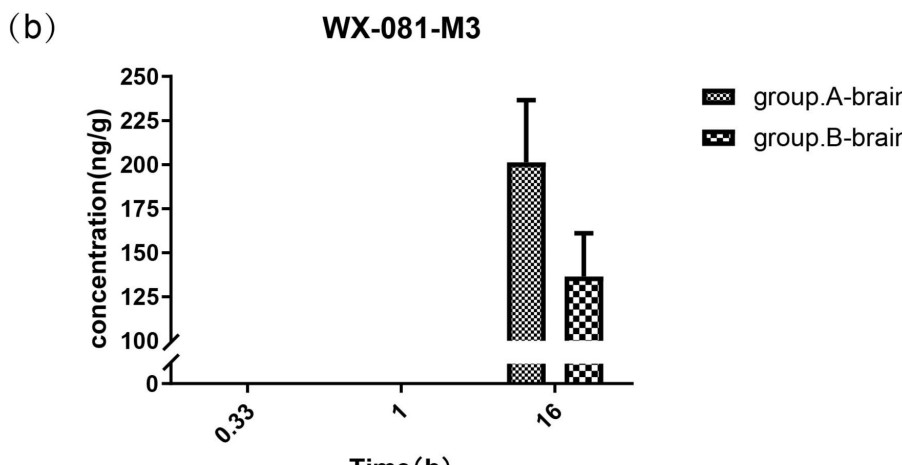

**FIG 2** Concentrations of WX-081 (a) and WX-081-M3 (b) in the brain tissue from groups A, B, and C at three time points. ns: not significant.

tissue in this study indicates that WX-081 is a promising novel therapeutic drug for patients with mycobacterial osteomyelitis.

In pharmacokinetic studies, rats are the preferred animal model. This is due to their physiological and metabolic similarities to humans. The activity of their hepatic metabolic enzymes, such as CYP450, closely resembles that in humans. This allows for an accurate prediction of drug metabolism. This study investigated the *in vivo* distribution of WX-081 in rat tissue. The highest WX-081 concentrations were observed in the lung, followed by plasma, vertebra, and brain in descending order. At 20 min, both groups B (45 mg/kg WX-081 plus 10 mg/kg Clr) and C (45 mg/kg WX-081 plus 25 mg/kg CFZ) had higher plasma WX-081 levels than Group A (45 mg/kg WX-081 alone). At 1 h, Group B showed a higher plasma WX-081 concentration but lower concentrations in the lung for both WX-081 and WX-081-M3 than Group A. Group C also had a lower lung WX-081 concentration than Group A at 1 h. In this study, we observed that at 20 min, the plasma concentration of WX-081 was significantly higher when combined with Clr or CFZ than when used alone. This result may be related to the absorption and distribution processes of the drugs in the body. In the early stages of drug treatment (20 min), combination therapy may have promoted the absorption of WX-081, elevating plasma concentration. However, this difference disappeared at subsequent time points (1 and 16 h) possibly due to dynamic changes in drug metabolism and distribution. Although no similar

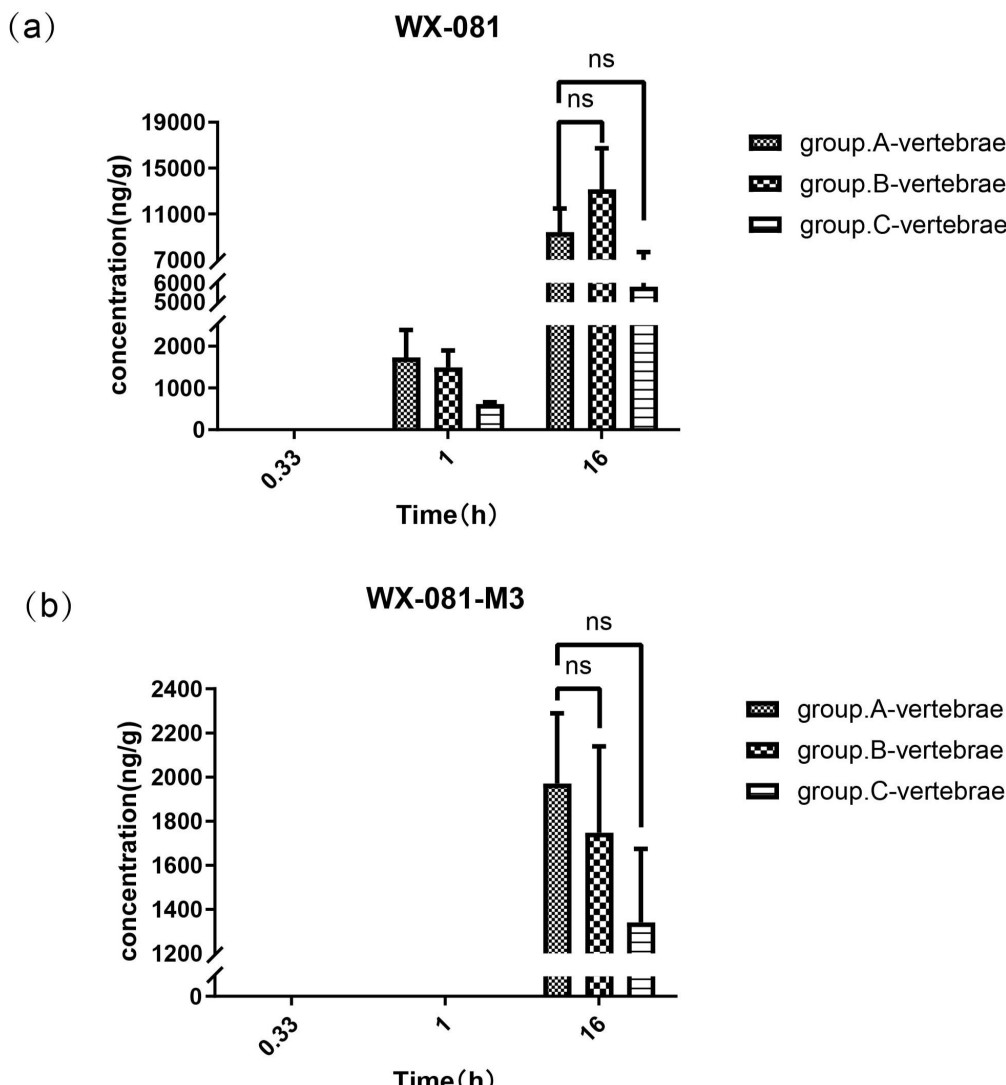

**FIG 3** Concentrations of WX-081 (a) and WX-081-M3 (b) in the vertebral tissue from groups A, B, and C at three time points. ns: not significant.

results were observed at other time points and in other tissue, the plasma concentration difference at 20 min remains significant. This indicates that in the early stages of drug treatment, combination therapy may affect the *in vivo* distribution of WX-081. Further-more, we noticed that at 1 h, the concentration of WX-081 in the lung tissue of Group B was higher than that of Group A. This further suggests that combination therapy may exert distinct drug distribution effects in specific tissue.

To the best of our knowledge, this study is the first to report drug-drug interactions between WX-081 and the currently administered anti-mycobacterial drugs Clr and CFZ. The combination use with clofazimine may potentially increase the plasma concentration of WX-081 during the absorption phase, yet it might decrease the concentration of WX-081 in the lung tissue at the time of reaching the peak concentration. Notably, an *in vitro* study demonstrated that a combination treatment of BDQ with CFZ showed greater therapeutic activity than BDQ alone (5), which is inconsistent with results obtained in the current *in vivo* study for WX-081, a BDQ analog. This discrepancy may be due to the fact that their study was performed *in vitro*, whereas our study was *in vivo*. Regardless, their study demonstrated lower therapeutic activity for BDQ combined with Clr than for BDQ alone, which contrasts with our results in this study showing higher WX-081 levels in

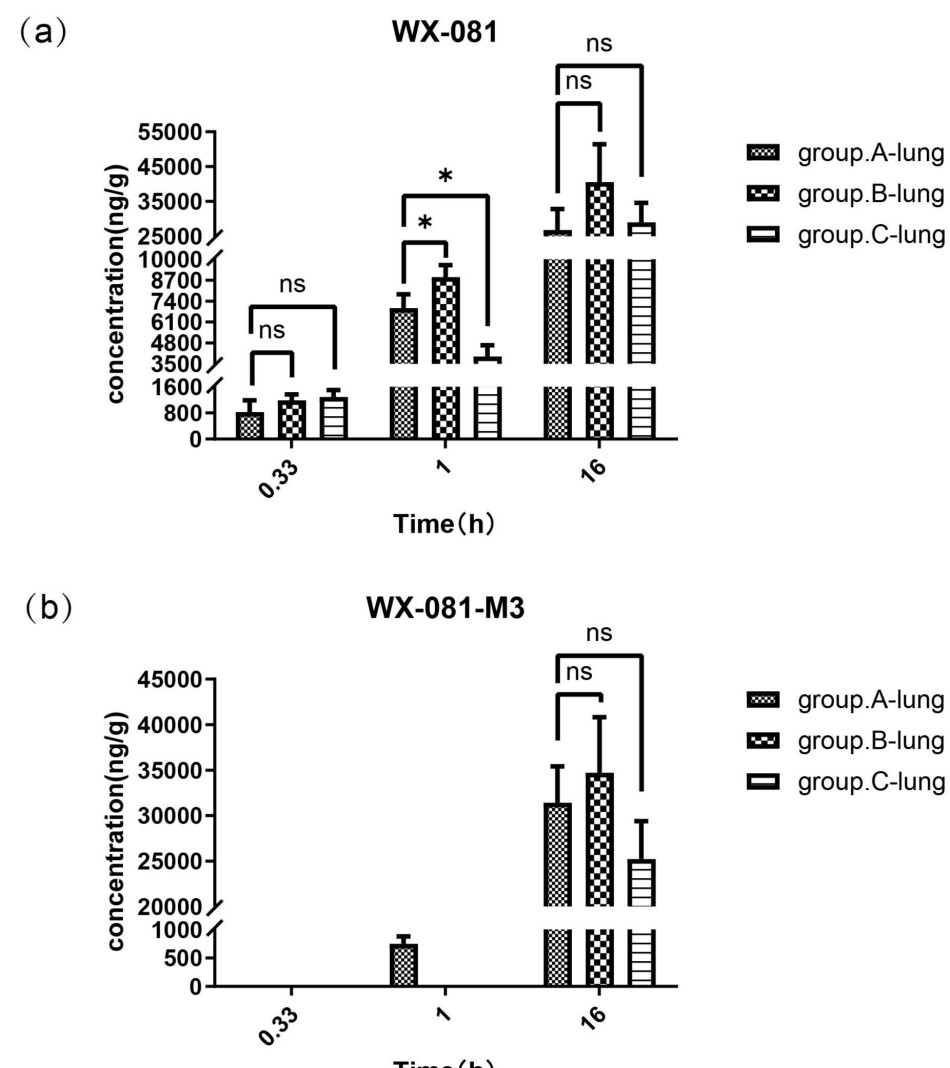

**FIG 4** Concentrations of WX-081 (a) and WX-081-M3 (b) in the lung tissue from groups A, B, and C at three time points. ns: not significant; *: $P < 0.05$.

Group B plasma, vertebrae, and lungs (but not brain tissue) at 1 h compared to the levels in Group A plasma, vertebrae, and lungs. And mechanistically, there might be three possible reasons. First, CFZ itself exhibits a high degree of accumulation in tissue, which may competitively inhibit the entry of BDQ into target tissue. Second, the reactive oxygen species induced by CFZ might conversely activate the bacterial stress response system. This leads to a decrease in the activity of the target of BDQ (ATP synthase) and a consequent weakening of the drug's effect. Third, exposure to CFZ may induce MTB to upregulate its efflux pump system, resulting in the efflux of BDQ and a subsequent decline in its intracellular concentration.

This study had two limitations. First, *in vivo* drug concentrations were only determined at three time points. As many time points as possible should have been included to ensure the comprehensiveness of the pharmacokinetic data obtained. Including more time points would enhance the comprehensiveness of the pharmacokinetic data. Due to methodological constraints, sequential plasma samples from individual animals could not be collected, preventing the calculation of conventional pharmacokinetic parameters. Nevertheless, the drug concentration data from multiple time points and tissues form a robust data set. This data set supports scientific conclusions on targeting efficacy, tissue accumulation tendency, and identification of potentially toxic organs. Second,

other tissues, such as the liver and the heart, should have been included in the analysis to obtain more data on drug metabolism and adverse effects. Addressing these limitations in future studies is crucial to validate our findings and further build upon the results of this study.

In conclusion, this study revealed that, at all time points examined, the highest concentrations of WX-081 across groups were found in the lung tissue. Compared to Group A, which received WX-081 (45 mg/kg) alone, both groups B (WX-081 plus Clr) and C (WX-081 plus CFZ) had higher plasma WX-081 concentrations at 20 min. At 1 h, Group B had higher plasma and lung tissue WX-081 concentrations and lower plasma concentration of the WX-081 metabolite WX-081-M3 than Group A. In contrast, Group C had lower lung tissue WX-081 concentration than Group A at 1 h post-treatment.

## MATERIALS AND METHODS

### Drugs

WX-081, WX-081-M3, Clr, and CFZ were purchased from Shanghai Jiatan Pharmatech Co., Ltd. (Shanghai, China). Loratadine (LLTD) was purchased from Shanghai McLean Biochemical Technology Co., Ltd. (Shanghai, China).

### Bacteria

The *M. abscessus* (ATCC 19977) were grown in Middlebrook 7H9 broth supplemented with 0.2% (vol/vol) glycerol, 0.05% Tween 80, and 10% (vol/vol) oleic acid-albumin-dextrose-catalase (Becton-Dickinson). Please refer to the relevant experimental steps in the reference for specific steps (11).

### Chromatographic conditions

LC-MS/MS was used to detect WX-081 and WX-081-M3 prototype compounds on an Agilent LC-MS/MS system (Agilent Technologies, Inc., Santa Clara, CA, USA) equipped with an Agilent XDB C18 column (50 mm × 3.5 mm, 2.1 µm). The LC-MS/MS system was operated using the following parameters: a flow rate 0.3 mL/min; mobile phase A of 0.1% formic acid solution and phase B of methanol; and injection volume of 5 µL. The gradient elution procedure used the following sequence: 0.0–0.3 min 90–10% B; 1.0–3.0 min 10–90% B; 3.1–4.0 min 90–10% B; 0.1–2.0 min 10% B.

### Mass spectrometry conditions

Positive electrospray ionization (ESI+) operating in multiple reaction monitoring (MRM) mode was used to collect signals associated with each compound. MS parameters were the following: drying gas temperature (gas temp) 300℃, drying gas flow rate (gas flow) 5 L/min, nebulizer pressure 45 PSI; sheath gas temperature 400℃, sheath gas flow rate 11 L/min, capillary voltage 3.5 kV, and nozzle voltage 500 V. Specific conditions used for LC-MS/MS detection of WX-081, its metabolite WX-081-M3, and the internal standard (ISTD) are listed in Table 2.

### Solution preparation

Ten milligrams each of WX-081 and WX-081-M3 was separately dissolved in 1 mL of dimethyl sulfoxide (DMSO) to obtain 10 mg/mL solutions of WX-081 and WX-081-M3 for use as LC/MS standards and stock solutions for experiments. Stock solutions were stored in the dark at low temperature. The ISTD stock solution was prepared by mixing 300 mL of acetonitrile and 15 µL of LLTD (100 µg/mL) in a mobile phase vial to obtain a 5 ng/mL LLTD solution.

### Establishment of the rat model and drug administration protocol

A total of 108 specific pathogen-free (SPF)-grade Crl:CD Sprague-Dawley (SD) rats of both sexes were used in this study. The age range of the rats was 6 to 8 weeks, and their

**TABLE 2** Detection conditions of drugs and IS LC-MS/MS

| Compound name | ISTD | Precursor | Product | Fragmentor | Collision | Polarity |
|---|---|---|---|---|---|---|
| WX-081 | | 537.1 | 310.2 | 150 | 26 | Positive |
| WX-081-M3 | | 524.5 | 463.5 | 135 | 15 | Positive |
| Loratadine | $\sqrt{}^a$ | 383 | 337.2 | 160 | 18 | Positive |

$^a\sqrt{}$, Loratadine was used as the internal standard (ISTD) in this LC-MS/MS experiment.

body weights ranged from 180 to 250 g. Rats received intravenous tail vein inoculation. Dexamethasone (5 mg/kg) in carboxymethyl cellulose was prepared fresh. Treatment started 1 week pre-infection, stopped 6 days post-infection, given daily, except on infection day. For lung implantation of 4.0–5.0 log$_{10}$CFU in mice, exponential-phase *M. abscessus* culture (A600nm 1.00–1.20) was diluted to A600nm 0.1 in saline. Please refer to the relevant experimental steps in the reference for specific steps (11). Rats were randomly divided into 18 groups (six animals per group consisting of three males and three females), which were combined to create groups A, B, and C, each comprising approximately 36 animals. Group A rats were administered WX-081 at 45 mg/kg; Group B rats were administered WX-081 at 45 mg/kg in combination with Clr at 10 mg/kg; and Group C rats were administered WX-081 at 45 mg/kg in combination with CFZ at 25 mg/kg. The dosages of the three drugs administered to rats were determined through conversion from clinical doses. All drugs were administered to the rats in the three groups by gavage at the same time. Animals were fasted for ≥12 h prior to drug administration. After drug administration, rats had *ad libitum* access to water, then 4 h later were given *ad libitum* access to both food and water. Euthanasia was performed by exsanguination via the femoral artery, and blood was collected at 20 min, 1 h, and 16 h after drug administration, followed by immediate harvesting of the tissue (brain, vertebral, and lung tissue). Both blood plasma and tissue samples were promptly frozen, and then stored at −80°C for use in subsequent experiments.

## Pre-treatment of plasma and tissue samples

About 0.1 g each of rat brain, vertebral, and lung tissue samples was separately homogenized in a 1:9 (v/v) saline:water solution. Subsequently, 50 µL of each tissue homogenate or plasma sample was transferred to separate 1.5 mL Eppendorf (EP) tubes. Then, after adding 150 µL of 5 ng/mL ISTD solution to each tube and vortexing to precipitate proteins, fourfold diluted samples were prepared. Similarly, after transferring 10 µL of each tissue homogenate or plasma sample to separate 1.5 mL EP tubes, 40 µL of blank matrix was added to each tube, followed by vortexing to prepare fivefold diluted samples. Then, to prepare 10-fold dilutions of the samples, 5 µL of each tissue homogenate or plasma sample was added to individual 1.5-mL EP tubes, followed by the addition of 45 µL of blank matrix and mixing by vortexing. All samples were centrifuged at 14,000 rpm for 10 min at 10°C, and then supernatants were extracted and analyzed by LC-MS/MS.

## Preparation of WX-081 and WX-081-M3 standard curves

To generate WX-081 and WX-081-M3 standard curves for plasma, vertebral tissue, and lung tissue, 47.5 µL of each tissue homogenate or plasma sample was transferred to separate 1.5-mL EP tubes, then 2.5 µL of WX-081 or WX-081-M3 at concentrations of 1, 2, 4, 10, 16, and 20 µg/mL was added to the tubes to obtain WX-081 and WX-081-M3 concentrations of 50, 100, 200, 500, 800, and 1,000 ng/mL, respectively.

For the preparation of the standard curve for brain tissue samples, 47.5 µL of brain homogenate was transferred to a 1.5 mL EP tube, then 2.5 µL of WX-081 or WX-081-M3 at concentrations of 0.2, 0.4, 1, 2, 4, 10, 16, and 20 µg/mL was added in parallel to generate standard curves for brain tissue samples of 10, 20, 50, 100, 200, 500, 800, and 1,000 ng/mL, respectively. All samples were prepared using the sample pre-treatment protocol described in the previous section.

The results were subjected to regression analysis using the weighted ($W = 1/X^2$) least squares method, with the concentration of WX-081 or WX-081-M3 serving as the horizontal coordinate and the peak area of the compound to be measured serving as the vertical coordinate. Linear regression equations were used to fit standard curves.

## Statistical analyses

All measurements were recorded and processed using the MassHunter software (Agilent Technologies, Inc.), then data were processed using Microsoft Office Excel 2010 (Microsoft Corporation, Redmond, WA, USA). Differences between continuous variables were analyzed using Student's $t$-test, with $P < 0.05$ considered statistically significant.

## ACKNOWLEDGMENTS

We thank the sample bank of Beijing Chest Hospital, Capital Medical University, Beijing Tuberculosis and Thoracic Tumor Research Institute, and Shanghai Jiatan Pharmatech Co., Ltd. for their help and support.

This study was funded by Beijing Municipal Health Commission Dengfeng Program (G202511062), Beijing Municipal Health Commission Excellent Clinical Research Program for Research-Oriented Wards (BRWEP2024W042160109), and Beijing Tongzhou District Science and Technology Program Project (WS2024017).

Xueyu Wang and Shan Gao contributed to the experimental operation, data curation, formal analysis, and manuscript writing; Xia Yu contributed to the data curation, validation, and manuscript writing; Yongguo Li and Lei Li contributed to the experimental operation; Naihui Chu contributed to the supervision and the revision of the manuscript; Hairong Huang contributed to the methodology and the revision of the manuscript; and Wenjuan Nie contributed to the funding acquisition, resources, supervision, and the revision of the manuscript. All authors contributed to the manuscript and approved the submitted version.

## AUTHOR AFFILIATIONS

[1]Tuberculosis Department, Beijing Tuberculosis and Thoracic Tumor Research Institute, Beijing Chest Hospital, Capital Medical University, Beijing, China
[2]Department of Ultrasound, Beijing Anzhen Hospital, Capital Medical University, Beijing, China
[3]National Clinical Laboratory on Tuberculosis, Beijing Key Laboratory for Drug Resistant Tuberculosis Research, Beijing Tuberculosis and Thoracic Tumor Institute, Beijing Chest Hospital, Capital Medical University, Beijing, China
[4]Shanghai Jiatan Pharmatech Co., Ltd., Shanghai, China

## AUTHOR ORCIDs

Xueyu Wang http://orcid.org/0009-0006-3684-3861
Xia Yu http://orcid.org/0000-0002-1400-7321
Hairong Huang http://orcid.org/0000-0002-4295-4262
Wenjuan Nie http://orcid.org/0000-0003-1109-1983

## FUNDING

| Funder | Grant(s) | Author(s) |
| --- | --- | --- |
| Beijing Municipal Health Commission Dengfeng Program | G202511062 | Wenjuan Nie |
| Beijing Municipal Health Commission Excellent Clinical Research Program for Research-Oriented Wards | BRWEP2024W042160109 | Wenjuan Nie |

| Funder | Grant(s) | Author(s) |
|---|---|---|
| Beijing Tongzhou District Science and Technology Program Project | WS2024017 | Wenjuan Nie |

## AUTHOR CONTRIBUTIONS

Xueyu Wang, Data curation, Formal analysis, Investigation, Writing – original draft | Shan Gao, Data curation, Formal analysis, Investigation, Writing – original draft | Xia Yu, Data curation, Validation, Writing – original draft | Yongguo Li, Investigation | Lei Li, Investigation | Naihui Chu, Supervision, Writing – review and editing | Hairong Huang, Methodology, Writing – review and editing | Wenjuan Nie, Funding acquisition, Resources, Supervision, Writing – review and editing

## DATA AVAILABILITY

All the data are available add and have already be published in the supplemental material.

## ETHICS APPROVAL

This study was approved by the Ethics Committee of Beijing Chest Hospital, Capital Medical University (Beijing, China) under protocol approval number 2021-020.

## ADDITIONAL FILES

The following material is available online.

### Supplemental Material

**Supplemental material (Spectrum01555-25-s0001.docx).** Fig. S1 and S2; Tables S1 and S2.

### Open Peer Review

**PEER REVIEW HISTORY (review-history.pdf).** An accounting of the reviewer comments and feedback.

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
