## [Reviewer comments · Microbiology Spectrum]

Microbiology Spectrum

Increasing *In Vivo* Drug Exposure Levels of Compound WX-081 (Sudapyridine) When Used in Combination with Clofazimine or Clarithromycin

Xueyu Wang, SHAN GAO, Xia Yu, Yongguo Li, Lei Li, Naihui Chu, Hairong Huang, and Wenjuan Nie

Corresponding Author(s): Wenjuan Nie, Beijing Chest Hospital affiliated to Capital Medical University

Review Timeline:

Submission Date:	May 18, 2025
Editorial Decision:	July 21, 2025
Revision Received:	October 6, 2025
Accepted:	October 21, 2025

Editor: Valeria Allizond

Reviewer(s): Disclosure of reviewer identity is with reference to reviewer comments included in decision letter(s). The following individuals involved in review of your submission have agreed to reveal their identity: Hao Li (Reviewer #1)

Transaction Report:

DOI: <https://doi.org/10.1128/spectrum.01555-25>

Re: Spectrum01555-25 (**Increasing *In Vivo* Drug Exposure Levels of Compound WX-081 (Sudapyridine) When Used in Combination with Clofazimine or Clarithromycin**)

Dear Dr. Wenjuan Nie:

Thank you for the privilege of reviewing your work. Below you will find my comments, instructions from the Spectrum editorial office, and the reviewer comments.

Revision Guidelines

Sincerely,
Valeria Allizond
Editor
Microbiology Spectrum

Reviewer #1 (Comments for the Author):

In this study, the authors used LC-MS/MS to investigate the in vivo distribution of WX-081 and with or without CLR or CFZ. This study gives important references for WX-081's research and clinical study.

Major concerns:

1 : The results indicate that only at 20mins, group B and group C had higher plasma WX-081 concentrations, which support the title. However, the results cannot be found at other times points as well as in other tissues.

2: The other pharmacokinetics parameters such as volume of distribution, Clearance, Pk/Pd, and Cmax/through, etc are not determined in this study.

Comments:

1 : Line70~71: Table S1 is the accuracy of LC-MS/MS determination of WX-081 and WX-081-M3. The authors need to check this sentence.

2: Figure 1 and Table 1 are the same datasets with different display models. Figure 1 can be as a supplementary figure. And in Figure 1, the concentration of plasma is ng/ml, while the others are ng/g. Therefore, it is better not display 2 units on the Y axis. The plasma results can be displayed in one figure.

3: The Figure legends of Figure 4, The marks should be Group A, Group B and Group C on Figure 4.

4: In the introduction or discussion part, the authors may point out why choose the rat as the animal model but not mouse, guinea pig and rabbit, etc.

5: In the discussion part, the authors may discuss or cite the references why choose the treatment concentrations of the three drugs for rats in this study.

6: The authors may indicate the age and weight ranges of the rates used in this study.

7: Since the rats are divided into males and females, did the authors compare the concentrations of the drugs in different tissues between males and female rats?

Reviewer #3 (Comments for the Author):

In this manuscript, Wang and Gao et al. examine the biological distribution of the diarylpyridine molecule WX-081 and its metabolite in a rodent model in the presence and absence of WHO recommended anti-NTM drugs such as clarithromycin and clofazimine. The authors methodically study drug distribution of WX-081, its metabolite WX-081-M3 in lung, brain and vertebral tissue as well as plasma- all crucial niches for the pathogen in NTM infections- to show that WX-081 distribution is highest in lung tissue but that the drug is also able to cross the blood-brain barrier and also reach vertebral tissue.

Overall, the methods of the study are well described and the authors also delineate the limitations of the study clearly by comparing their findings with those reported previously in literature. However, the data representation in the manuscript is unclear which makes it difficult to assess the robustness of the conclusions drawn. My concerns with the manuscript are listed below:

Major

- All graphs in Figure 1 and figures 2B, 3A-B, 4A-B, and 5B need to be rescaled on the Y-axis to make the differences clearer. The data is hard to understand in the current format of the plots.
- Fig. 1A- "organisation distribution" is unclear. Do the authors mean distribution of WX-081 and its metabolite in various organs? Additionally, the units on the Y-axis are unclear and should be explained in the figure legend text.
- In figures 2-5, there seems to be no significant differences in the distribution of WX-081 or its metabolite in plasma, brain, lung or vertebral tissue at 16h while the data points for earlier timepoints in most cases are absent/unclear. Based on the data presented here, it is premature to make any conclusions about any drug-drug interactions between WX-081 and clarithromycin/clofazimine. Did the authors use the mean concentrations reported in Table 1 as the basis for their conclusions? The inferences drawn in lines 42-51 based on these numbers seems speculative at best.
- How would the authors try to mechanistically reconcile the differences in drug-drug interactions as reported in vitro in a previous study with bedaquiline, the analogue of WX-081 (PMID: 30649327)
- Particularly in the case of clofazimine, which would disrupt redox cycling to interfere with the electron transport chain and compromise energy metabolism like bedaquiline and its analogues, how do the authors explain no significant interactions in their combination studies?

Minor

- The manuscript would benefit in readability through use of shorter sentences to make the information flow more lucid.

In this manuscript, Wang and Gao et al. examine the biological distribution of the diarylpyridine molecule WX-081 and its metabolite in a rodent model in the presence and absence of WHO recommended anti-NTM drugs such as clarithromycin and clofazimine. The authors methodically study drug distribution of WX-081, its metabolite WX-081-M3 in lung, brain and vertebral tissue as well as plasma- all crucial niches for the pathogen in NTM infections- to show that WX-081 distribution is highest in lung tissue but that the drug is also able to cross the blood-brain barrier and also reach vertebral tissue.

Overall, the methods of the study are well described and the authors also delineate the limitations of the study clearly by comparing their findings with those reported previously in literature. However, the data representation in the manuscript is unclear which makes it difficult to assess the robustness of the conclusions drawn. My concerns with the manuscript are listed below:

Major

- All graphs in Figure 1 and figures 2B, 3A-B, 4A-B, and 5B need to be rescaled on the Y-axis to make the differences clearer. The data is hard to understand in the current format of the plots.
- Fig. 1A- "organisation distribution" is unclear. Do the authors mean distribution of WX-081 and its metabolite in various organs? Additionally, the units on the Y-axis are unclear and should be explained in the figure legend text.
- In figures 2-5, there seems to be no significant differences in the distribution of WX-081 or its metabolite in plasma, brain, lung or vertebral tissue at 16h while the data points for earlier timepoints in most cases are absent/unclear. Based on the data presented here, it is premature to make any conclusions about any drug-drug interactions between WX-081 and clarithromycin/clofazimine. Did the authors use the mean concentrations reported in Table 1 as the basis for their conclusions? The inferences drawn in lines 42-51 based on these numbers seems speculative at best.
- How would the authors try to mechanistically reconcile the differences in drug-drug interactions as reported in vitro in a previous study with bedaquiline, the analogue of WX-081 (PMID: 30649327)
- Particularly in the case of clofazimine, which would disrupt redox cycling to interfere with the electron transport chain and compromise energy metabolism like bedaquiline and its analogues, how do the authors explain no significant interactions in their combination studies?

Minor

- The manuscript would benefit in readability through use of shorter sentences to make the information flow more lucid.

Comment to editor:

The study lacks clarity in the presentation of results, which makes it difficult to assess the robustness of the data and relevance of the conclusions drawn. In the current version, the inferences drawn about drug-drug interactions of WX-081 with clarithromycin/clofazimine seem speculative at best. The manuscript in its current format is unsuitable for acceptance- I would recommend that the authors be asked to modify the manuscript significantly to add robustness to their conclusions.

Reviewer #1 (Comments for the Author):

In this study, the authors used LC-MS/MS to investigate the in vivo distribution of WX-081 and with or without CLR or CFZ. This study gives important references for WX-081's research and clinical study.

Major concerns:

1: The results indicate that only at 20mins, group B and group C had higher plasma WX-081 concentrations, which support the title. However, the results cannot be found at other times points as well as in other tissues.

Reply: Thank you for your valuable feedback. The results you pointed out show that the plasma WX-081 concentration in Group B and Group C is higher than that in Group A only at 20 minutes, which is consistent with the title. However, similar results were not found at other time points and in other tissues, which indeed requires further discussion.

Based on the editor's comments, we have incorporated "In this study, we observed that at 20 min, the plasma concentration of WX-081 was significantly higher when combined with Clr or CFZ than when used alone. This result may be related to the absorption and distribution processes of the drugs in the body. In the early stages of drug treatment (20 min), combination therapy may have promoted the absorption of WX-081, elevating plasma concentration. However, this difference disappeared at subsequent time points (1 h and 16 h), possibly due to dynamic changes in drug metabolism and distribution. Although no similar results were observed at other time points and in other tissue, the plasma concentration difference at 20 min remains significant. This indicates that in the early stages of drug treatment, combination therapy may affect the in vivo distribution of WX-081. Furthermore, we noticed that at 1 h, the concentration of WX-081 in lung tissue of Group B was higher than that of Group A. This further suggests that combination therapy may exert distinct drug distribution effects in specific tissue." into the 9th to 19th line in the 3rd paragraph of the "**Discussion**" part.

We understand the editor's concern regarding the consistency between the title and the results. We believe that, despite variations in results across different time points and tissues, the title still accurately reflects the impact of combination therapy on the in vivo distribution of WX-081 observed at 20 minutes.

2: The other pharmacokinetics parameters such as volume of distribution, Clearance, Pk/Pd, and Cmax/through, etc are not determined in this study.

Reply: Thank you for highlighting the absence of certain pharmacokinetic parameters. Parameters such as distribution volume, clearance rate, PK/PD ratios, and Cmax/through are indeed vital for a comprehensive pharmacokinetic evaluation. We have also incorporated an introduction to the pharmacokinetic parameters of WX-081 into 7th to 14th line in the 1st paragraph of the "**Discussion**" part.

However, the primary aim of this research is to describe the tissue distribution pattern of WX-081 in rats. This will provide direct evidence needed to evaluate its targeting efficacy and safety. The experimental design entailed the euthanasia of rats at specific time intervals for tissue collection. The main objective was to obtain accurate drug concentration measurements in various tissues. Owing to the intrinsic limitations of this methodology, we could not collect sequential plasma samples from individual animals, consequently precluding the calculation of conventional

pharmacokinetic parameters. Nevertheless, the drug concentration data gathered from multiple time points and tissues in this study form a robust and self-contained dataset. This dataset sufficiently supports scientific conclusions about the drug's targeting efficacy, tendency to accumulate in tissues, and identification of potentially toxic organs. However, we acknowledge the necessity of measuring additional parameters for a comprehensive understanding of WX-081's pharmacokinetics. PK/PD assessments will also be conducted to optimize dosing and support clinical applications.

Based on the editor's comments, we have incorporated "Including more time points would enhance the comprehensiveness of the pharmacokinetic data. Due to methodological constraints, sequential plasma samples from individual animals could not be collected, preventing the calculation of conventional pharmacokinetic parameters. Nevertheless, the drug concentration data from multiple time points and tissues form a robust dataset. This dataset supports scientific conclusions on targeting efficacy, tissue accumulation tendency, and identification of potentially toxic organs." into the 3rd to 8th line in the 5th paragraph of the "Discussion" part.

Comments:

1: Line70~71: Table S1 is the accuracy of LC-MS/MS determination of WX-081 and WX-081-M3. The authors need to check this sentence.

Reply: Thank you for your advice. We have changed the title of Table S1 to "Concentration determination and accuracy analysis of WX-081 and WX-081-M3 in plasma, brain, vertebrae, and lung tissue."

2: Figure 1 and Table 1 are the same datasets with different display models. Figure 1 can be as a supplementary figure. And in Figure 1, the concentration of plasma is ng/ml, while the others are ng/g. Therefore, it is better not display 2 units on the Y axis. The plasma results can be displayed in one figure.

Reply: Thank you for your advice. We have renamed Figure 1 as Figure S2 and placed it in the supplementary material, with the sequence of the other figures shifted forward accordingly. Additionally, we have specified in the caption of Figure S2 that the unit for plasma concentration is ng/ml, while the units for other tissues are ng/g. Presenting the plasma results in one figure might lead to some overlap with the content displayed in the new Figure 1 and would fail to highlight the conclusion that "Lung tissue consistently had the highest WX-081 concentration at all time points." Therefore, we still believe it is more appropriate to present the plasma results together with the results of other tissues in a single figure.

3: The Figure legends of Figure 4, The marks should be Group A, Group B and Group C on Figure 4.

Reply: Thank you for your advice. It's our mistakes. We've made modifications in accordance with your requirements in Figure 4 (now, it is Figure 3).

4: In the introduction or discussion part, the authors may point out why choose the rat as the animal model but not mouse, guinea pig and rabbit, etc.

Reply: Thank you for your advice. In pharmacokinetic studies, rats are the preferred animal model. This is due to their physiological and metabolic similarities to humans. The activity of their hepatic metabolic enzymes, such as CYP450, closely resembles that in humans. This allows for accurate prediction of drug metabolism. In contrast, mice exhibit an accelerated metabolic rate, potentially leading to underestimation of drug half-life. The moderate size of rats enables multiple

blood samplings for constructing pharmacokinetic profiles. Mice, however, are less suitable for serial sampling due to their small size. Guinea pigs show significant differences from humans in certain metabolic pathways, such as acetylation. Rabbits, as herbivores, may exhibit altered drug absorption and distribution. Rats facilitate integration with toxicological and pharmacodynamic studies, ensuring data consistency. They also offer cost and ethical advantages. Thus, considering metabolic similarity, experimental feasibility, and research integration, rats are the optimal rodent model for pharmacokinetic studies. Mice, guinea pigs, and rabbits are better suited for specific research purposes.

Based on the editor's comments, we've incorporated "In pharmacokinetic studies, rats are the preferred animal model. This is due to their physiological and metabolic similarities to humans. The activity of their hepatic metabolic enzymes, such as CYP450, closely resembles that in humans. This allows for accurate prediction of drug metabolism." into the 1st to 3rd line in the 3rd paragraph of the "**Discussion**" part.

5: In the discussion part, the authors may discuss or cite the references why choose the treatment concentrations of the three drugs for rats in this study.

Reply: Thank you for your advice. We've incorporated "The dosages of the three drugs administered to rats were determined through conversion from clinical doses." into the 11th to 12th line of the "**Establishment of rat model and drug administration protocol**" part.

6: The authors may indicate the age and weight ranges of the rats used in this study.

Reply: Thank you for your advice. We've incorporated "The age range of the rats was 6 to 8 weeks. And their body weights ranged from 180 to 250 g." into the second line of the "**Establishment of rat model and drug administration protocol**" part.

7: Since the rats are divided into males and females, did the authors compare the concentrations of the drugs in different tissues between males and female rats?

Reply: Thank you for your advice. We did not conduct a direct comparison of drug concentrations in different tissues between male and female rats. The primary reason for this decision is that our research was initially designed to focus on the overall tissue distribution characteristics of the drugs in rats as a general model, rather than specifically exploring gender-related differences. In conclusion, we prioritized obtaining comprehensive data on drug concentrations across multiple tissues in a combined male-female rat population to ensure the robustness of our overall findings. Furthermore, there was no difference in concentration and pharmacokinetics between males and females in the Phase 1 and Phase 2 clinical trials of this drug, hence no distinction was made between the sexes.

Reviewer #3 (Comments for the Author):

In this manuscript, Wang and Gao et al. examine the biological distribution of the diarylpyridine molecule WX-081 and its metabolite in a rodent model in the presence and absence of WHO recommended anti-NTM drugs such as clarithromycin and clofazimine. The authors methodically study drug distribution of WX-081, its metabolite WX-081-M3 in lung, brain and vertebral tissue as well as plasma- all crucial niches for the pathogen in NTM infections- to show that WX-081 distribution is highest in lung tissue but that the drug is also able to cross the blood-brain barrier and also reach vertebral tissue.

Overall, the methods of the study are well described and the authors also delineate the limitations of the study clearly by comparing their findings with those reported previously in

literature. However, the data representation in the manuscript is unclear which makes it difficult to assess the robustness of the conclusions drawn. My concerns with the manuscript are listed below:

Major

- All graphs in Figure 1 and figures 2B, 3A-B, 4A-B, and 5B need to be rescaled on the Y-axis to make the differences clearer. The data is hard to understand in the current format of the plots.

Reply: Thank you for your advice. We have completed the modifications to these figures (now, they are Figure 2S, Figure 1-4).

- Fig. 1A- "organisation distribution" is unclear. Do the authors mean distribution of WX-081 and its metabolite in various organs? Additionally, the units on the Y-axis are unclear and should be explained in the figure legend text.

Reply: Thank you for your advice. "Organisation distribution" means the concentration of WX-081 and WX-081-M3 in plasma and other tissue (brain, vertebrae and lung). Additionally, we have specified in the caption of Figure 1 (now, it is Figure S2) that the unit for plasma concentration is ng/ml, while the units for other tissues are ng/g.

- In figures 2-5, there seems to be no significant differences in the distribution of WX-081 or its metabolite in plasma, brain, lung or vertebral tissue at 16h while the data points for earlier timepoints in most cases are absent/unclear. Based on the data presented here, it is premature to make any conclusions about any drug-drug interactions between WX-081 and clarithromycin/clofazimine. Did the authors use the mean concentrations reported in Table 1 as the basis for their conclusions? The inferences drawn in lines 42-51 based on these numbers seems speculative at best.

Reply: Thank you for your advice. We agree with this assessment. Instead of stating them as definite findings, we now present them as speculative inferences on the 2nd to 4th line in the 4th paragraph of the "Discussion" part as followed: "The combination use with clofazimine may potentially increase the plasma concentration of WX-081 during the absorption phase, yet it might decrease the concentration of WX-081 in lung tissue at the time of reaching the peak concentration."

- How would the authors try to mechanistically reconcile the differences in drug-drug interactions as reported in vitro in a previous study with bedaquiline, the analogue of WX-081 (PMID: 30649327)

Reply: Thank you for your advice. This discrepancy may be due to the fact that their study was performed in vitro, whereas our study was in vivo. We have mentioned this in the 7th to 8th line in the 4th paragraph of the "Discussion" part. And we have incorporated "And mechanistically there might be three possible reasons. Firstly, CFZ itself exhibits a high degree of accumulation in tissue, which may competitively inhibit the entry of BDQ into target tissue. Secondly, the reactive oxygen species induced by CFZ might conversely activate the bacterial stress response system. This leads to a decrease in the activity of the target of BDQ (ATP synthase) and a consequent weakening of the drug's effect. Thirdly, exposure to CFZ may induce MTB to upregulate its efflux pump system, resulting in the efflux of BDQ and a subsequent decline in its intracellular concentration." into the 11th to 16th line in the 4th paragraph of the "Discussion" part.

- Particularly in the case of clofazimine, which would disrupt redox cycling to interfere with

the electron transport chain and compromise energy metabolism like bedaquiline and its analogues, how do the authors explain no significant interactions in their combination studies?

Reply: Thank you for your advice. The meaning that this question intends to convey appears to be identical to that of the previous one. Based on the editor's comments, we have incorporated “And mechanistically there might be three possible reasons. Firstly, CFZ itself exhibits a high degree of accumulation in tissue, which may competitively inhibit the entry of BDQ into target tissue. Secondly, the reactive oxygen species induced by CFZ might conversely activate the bacterial stress response system. This leads to a decrease in the activity of the target of BDQ (ATP synthase) and a consequent weakening of the drug's effect. Thirdly, exposure to CFZ may induce MTB to upregulate its efflux pump system, resulting in the efflux of BDQ and a subsequent decline in its intracellular concentration.” into the 11th to 16th line in the 4th paragraph of the “**Discussion**” part.

Minor

• The manuscript would benefit in readability through use of shorter sentences to make the information flow more lucid.

Reply: Thank you for your advice. We have already engaged native English-speaking scholars to polish the language of the article.

Correction(s) needed

1. Please include a list of figure legends at the end of the manuscript, after the references section.

Reply: Thank you for your advice. We have added the figure legends after the references section.

2. The Importance paragraph is currently missing from the Manuscript Text File. The Importance section (150 words or shorter) is a nontechnical explanation of the significance of the study to the field. It should be inserted immediately after the Abstract in the text file (i.e., not just in the submission form) and labelled as 'Importance.' Please add this to your text file.

Reply: Thank you for your advice. We have added the importance paragraph after the abstract in the text file.

3. Please provide a compare PDF file that indicates the changes from the original submission (by highlighting or underlining the changes). Please upload this PDF file separately from the manuscript file as file type "Marked Up Manuscript - For Review Only File."

Reply: Thank you for your advice. We have uploaded a PDF file that indicates the changes from the original submission by highlighting the changes.

Re: Spectrum01555-25R1 (**Increasing *In Vivo* Drug Exposure Levels of Compound WX-081 (Sudapyridine) When Used in Combination with Clofazimine or Clarithromycin**)

Dear Dr. Wenjuan Nie:

Your manuscript has been accepted, and I am forwarding it to the ASM production staff for publication. Your paper will first be checked to make sure all elements meet the technical requirements. ASM staff will contact you if anything needs to be revised before copyediting and production can begin. Otherwise, you will be notified when your proofs are ready to be viewed.

Sincerely,
Valeria Allizond
Editor
Microbiology Spectrum

Reviewer #1 (Comments for the Author):

The questions and suggestions are well answered by the authors.

Reviewer #3 (Comments for the Author):

The authors have addressed all major and minor concerns raised.

The authors have addressed all major and minor concerns raised.